# Feasibility Study for Using Piezoelectric-Based Weigh-In-Motion (WIM) System on Public Roadway

**Haocheng Xiong [1] and Yinning Zhang [2],***

[1]   National Center for Materials Service Safety, University of Science and Technology Beijing,
     Beijing 100083, China
[2]   Department of Civil & Resources Engineering, University of Science and Technology Beijing,
     Beijing 100083, China
*   Correspondence: hcxiong@vt.edu; Tel.: +86-13071130135

**Abstract:** Weigh-in-Motion system has been the primary selection of U.S. government agencies as the weighing enforcement for decades to protect the road pavement. In recent years, the number of trucks has increased by about 40% and in 2017, they travel 25% more annually than in 2016. The lack of the budget has slowed down the expansion of weighing enforcement to catch up with the growing workload of vehicle weighing. Unsupervised pavement section suffers more pavement damage and increased repairing cost. In this work, a piezoelectric material based WIM system (P-WIM) is developed. Such a system consists of several piezoelectric material disks that are capable of generating characteristic voltage output from passing vehicles. The axle loading of the vehicle can be determined by analyzing the voltage generated from the P-WIM. Compared to traditional WIM system, P-WIM requires nearly zero maintenance and costs 80% less on capital investment and less labor and effort to integrate. To evaluate the feasibility of this technology to serve as weighing enforcement on public roadways, prototype P-WIMs are fabricated and installed at a weigh station. The vehicle loading information provided by the weigh station is used to determine the force transmission percentage of the installed P-WIMs, which is an important parameter to determine the vehicles' axle loading by generated voltage.

**Keywords:** Weigh-in-Motion; Piezoelectricity; Pavement Performance; Pavement Monitoring

## 1. Introduction

### 1.1. General Introduction

The over-loaded trucks have been a critical issue in many countries, which causes more damage to the pavement and reduces the service life of the roadway. Compared with the static weighing system, weigh-in-motion (WIM) system provides the traffic loading data with less accuracy, but much less impact on the traffic flow [1]. The WIM systems weigh vehicles at highway speeds and introduce much less delay to the transportation network than the static one. For these reasons, WIM systems have been widely deployed to support transportation infrastructure management [2–4]. It also serves as primary weight compliance checkpoints of commercial trucks for weight enforcement [5,6].

The total number of trucks in the United States has increased from 60 million to 85 million from 1992 to 2002 [7]. The annual travel mileage of heavy trucks has increased from 99.5 million to 138.6 million miles during the same period [8]. The increased workload promoted the deployment of additional WIM systems and weighing equipment. Installing a permanent WIM system in the pavement can be costly (about $250,000 for a four-lane road [9]. It brings a dilemma to government agencies that either increasing the budget to install more WIM systems by cutting off other projects or

tolerating the results of lacking weighing enforcement for today's increasing traffic. Highway Statistic 2017 indicates that the annual miles traveled by vehicles on rural roads decrease slightly from 2007 to 2017 (1,032,790 million miles to 963,206 million miles) [10]. However, according to the chapter "Growth in Volume and Loadings on the Rural Interstate System" included in the report Highway Statistic 2013 by Federal Highway Administration, the average daily load on rural roads is increasing annually since 1970 [11]. The average daily load on rural roads in 2013 is almost 6 times greater than in 1970. It indicates that there is a greater chance of overweight violation on rural roads today, which significantly reduces their service life. Researchers are developing new approaches to find alternative methods of weighing vehicles. However, in recent years, most of the development of WIM is conducted for monitoring the traffic over bridges. Chen et al. have developed a weigh-in-motion method to identify the vehicle speed and axle loading on bridges using long-gauge fibre Bragg grating (LGFBG) sensor [12]. Deng et al. proposed an equivalent shear force method to identify vehicle information, including speed and axle loading [13]. The research activities on WIM in the pavement were mostly focused on optimizing the existing system to achieve better accuracy or better understanding of the pavement-WIM interaction. Zhang et al. provided an optimization framework for the in-pavement WIM system in their study [14]. Burnos and Rys have conducted a comprehensive study to find the weighing accuracy of in-pavement WIM over the pavement/sensor complex behavior [15].

This paper presented a proof-of-concept evaluation of an alternative to permanent WIM system with much lower cost using the piezoelectric material, which is named as P-WIM. The P-WIM consists of several piezoelectric disks sealed in a protective package made from engineering plastic. The fabricated prototype P-WIMs are installed at the entrance of Troutville Weigh Station near Roanoke, VA. On-site evaluations are performed to find if the installed P-WIMs are capable of capturing the axle loading of passing vehicles. The axle loading data, measured by the Virginia Department of Motor Vehicle (VDMV) using their static weight, was used to find the relationship between the voltage generated from P-WIMs and the actual traffic loading.

## 1.2. Piezoelectricity

There are numbers of electric dipoles in every piezoelectric material. These electric dipoles will be aligned to a specific direction after a certain electric field is applied. The direction that electric dipoles aligned to are called the poling direction of the piezoelectric material. When a stress is applied along the poling direction of the piezoelectric material, the displacement of inner electric dipoles will create an electric potential between the two poles of the material. When the two poles are connected to a loading outside, the electric energy stored in the material is released. This phenomenon of converting mechanical energy into electric energy is called piezoelectricity. While the rapid development of the research of the mechanics of asphalt pavement and performance evaluations, piezoelectricity has been widely employed to establish the relation between moving vehicles and pavement deformation [16,17].

Piezoelectric material has been utilized in different applications, such as condition monitoring [18] and evaluating the structural health of the pavement [19–21]. Piezoelectric material has shown that it is very sensitive and responsive to external force. This characteristic makes it a great option for designing a load-measuring device. In most of the in-pavement WIM technologies, piezeceramic bars are used to measure the vehicle's weight [12]. After being developed for decades, piezoelectric materials have been fabricated into different shapes, with different compositions and properties. Piezoceramic lead zirconate titanate, known as PZT, is widely used in many designs [22] simply because it has the best cost-effectiveness among all kinds of piezoelectric materials. As a brittle material, PZT has low strain endurance, and its application is limited. C.S. Lee et al. have developed a Poly Vinylidene Fluoride (PVDF) film in their work that is coated with poly (3,4-ethylenedioxy-thiophene)/poly(4-styrenesulfonate) [PEDOT/PSS] electrodes [23]. With great flexibility, PVDF is capable of enduring large strains.

## 2. Methods

### 2.1. Coupling Modes

In different applications, piezoelectric materials convert mechanical energy into electric energy with different coupling modes. 33-mode and 31-mode are used in most of the piezoelectric applications. The concepts of 33-mode (Figure 1a) and 31-mode (Figure 1b) introduced by Anton, S. and Sodano, H. [24] are illustrated in Figure 1. Under 31-mode, the electrical output is maximized when the external forces are vertical to piezoelectric material's poling direction. Oppositely, under 33-mode, the electrical output from the piezoelectric material is maximized when it is stressed along its poling direction. Many researchers have done study and test about piezoelectric material's coupling modes [25–28]. They have concluded that in a traffic monitoring application, which employs piezoelectric material working under 31-mode, the sensitivity to the external force depends on the vehicle speed. It means that, if the vehicle traveling faster or slower than a certain range of speed (depending on the design of the device), the sensitivity of the device will be much lower. Since the electrical output depends mostly on the applied external force, the sensitivity to applied loading of the device designed to work under 33-mode is uniform regardless of the vehicle speed. For these reasons, the P-WIM is designed to work under 33-mode.

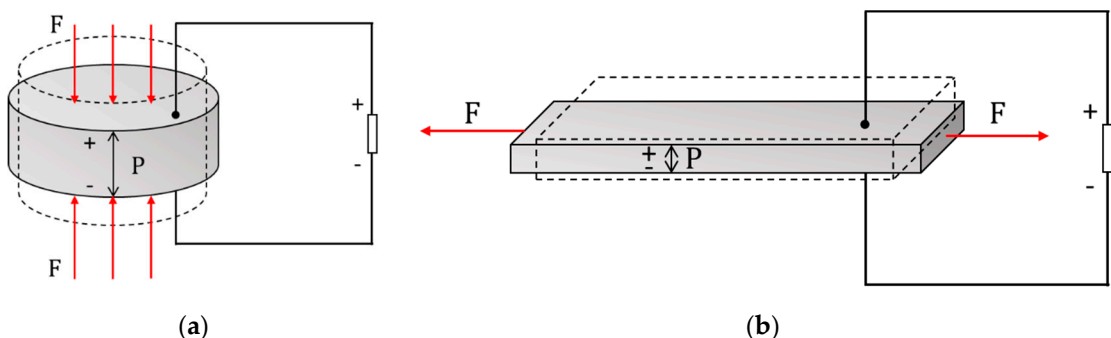

(**a**)　　　　　　　　　　　　　　　　　　　(**b**)

**Figure 1.** The illustration of 33- (**a**) and 31- (**b**) coupling mode of piezoelectric energy harvesting.

### 2.2. Material Selection

It is very important to select the piezoelectric material with material properties suitable for measuring the vehicles' axle loading. In many studies, the piezoelectric charge constant and the piezoelectric voltage constant has been approved to be highly related to the measurement accuracy [24,25,29]. In 33-mode, these two material properties are express as $d_{33}$ and $g_{33}$, which is directly related to the current and voltage output under the vehicles' load. In these studies, piezoelectric materials with lower elastic compliance are recommended due to their better electrical output performance under the same load. The quality factor, Q, has been investigated in Richards et al.'s study as another important factor to the error and variance of the measuring results [30]. During the measuring process, the piezoelectric material will endure less damping with a higher quality factor. As a conclusion, P-WIMs are fabricated using a piezoelectric material with higher piezoelectric coefficients, lower elastic compliance, and higher quality factor.

### 2.3. Fabrication

To prove that the P-WIM is feasible of working properly under realistic conditions, performing the on-site evaluation with prototype P-WIMs is necessary. To achieve the objectives, four prototype P-WIMs are fabricated in the laboratory. For each P-WIM, nine piezoelectric disks (Disk-shaped material is used to eliminate the stress concentration when force is applied) are sealed in a protective package made from engineering plastic, which is insulating and able to resist the external impact from vehicles with a long lifespan. Another function of this package is keeping fluid or other chemical

and contaminants out from the piezoelectric materials, which prevents any form of the short circuit of the electric circuit caused by fluid. Figure 2a shows a conceptual sketch of the prototype P-WIM. The cover part of the protective package is transmitting the load applied by vehicles uniformly onto the piezoelectric material. Ununiformed stress distribution on the piezoelectric material will significantly reduce their service life since they are naturally brittle (Some of the prototype P-WIMs failed quickly in lab tests because the force was not uniformly distributed onto the piezoelectric materials). Figure 2b shows a prototype P-WIM without a cover. All the positive and negative poles of the piezoelectric material are connected by several bar-shaped copper electrodes integrated into the protective package. The 850 series PZT piezoceramic disks are selected based on the criterions mentioned previously and in the material selection section. The piezoelectric charge constant, $d_{33}$, of the 850 series piezoceramic is $400 \times 10^{-12}$ C/N, the piezoelectric voltage constant; $g_{33}$, is $24.8 \times 10^{-3}$ Vm/N, and Young's modulus along 33 directions is $5.4 \times 10^{10}$ N/m$^2$.

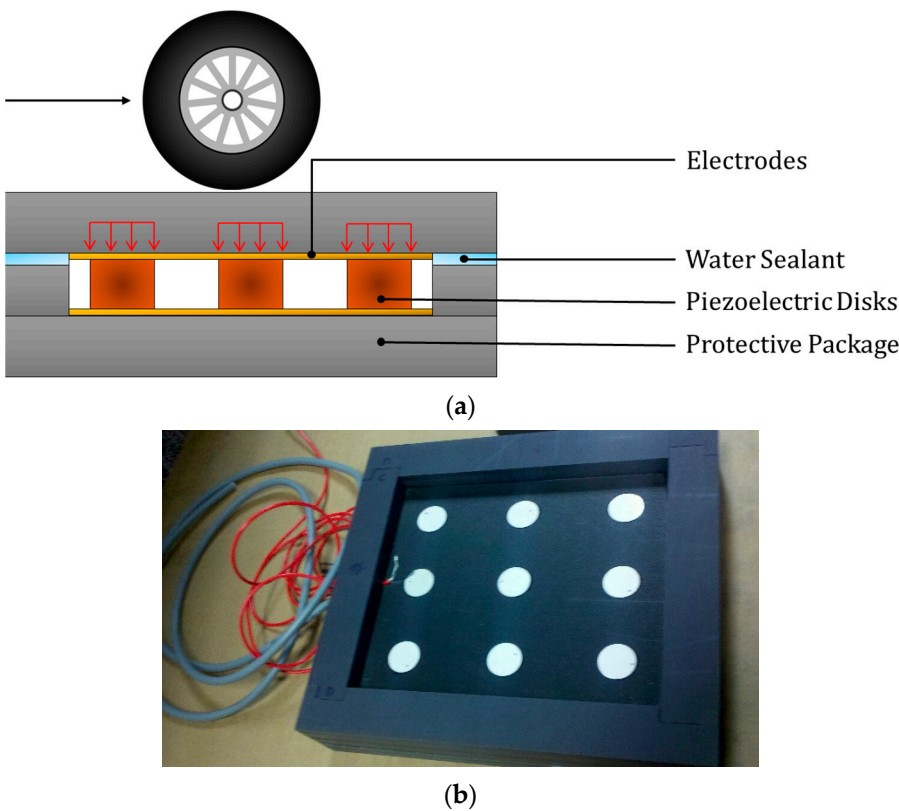

**(a)**

**(b)**

**Figure 2.** The prototype P-WIM (**a**) and the conceptual sketch (**b**).

### 2.4. Installation of the P-WIMs

To find if the P-WIM is capable of providing legitimate axle loading information of vehicle under the realistic condition, four fabricated prototype P-WIMs were installed at Troutville Weigh Station near Roanoke, VA. There are over 4000 trailer trucks passing the weigh station every day. The P-WIMs were installed at the weigh lane of the weigh station. The axle loading information of every vehicle passed through weigh lane will be recorded by VDMV. The data collected by VDMV is used to investigate the relationship between the vehicle axle loading and the generated voltage from P-WIMs.

The installation spacing of P-WIMs is determined from the tire configuration of regular trailer trucks. The P-WIMs were installed along the wheel paths of the vehicles to have better chance to have full contact with the tires of vehicles. The layout of the installation is illustrated in Figure 3. Four P-WIMs are installed in two rows on both sides of the cable channel. The tiles with a darker color in the figures represent the P-WIMs, and the numbers are the identification of each P-WIM.

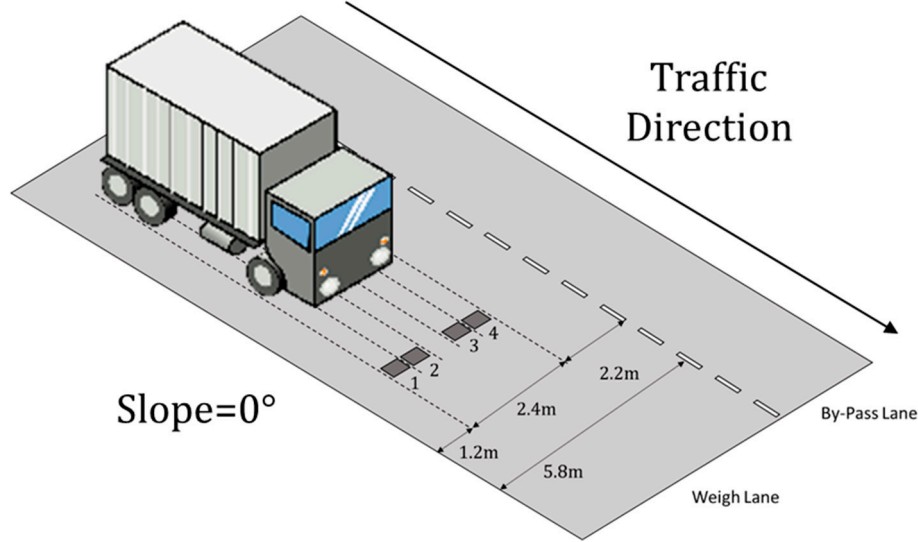

**Figure 3.** The layout of the installation of P-WIMs.

Before installing the P-WIMs, square-shaped pits were cut on the pavement (Figure 4b). Then the bottoms of the pits are leveled parallel to the surface of the pavement using cement grout to have a uniform load distribution on the base of the P-WIMs. As suggested by VDOT, engineering epoxy is used to seal the space in pits and keep the P-WIMs steady in the pavement. A metal plated is placed on top of each P-WIM, and the top surface of the metal plate is precisely aligned to the surface of the lane to avoid any potential impact from passing vehicles. Figure 4a is a sketch of the side view of the installation, and Figure 4c is a picture showing the installed P-WIMs under real traffic.

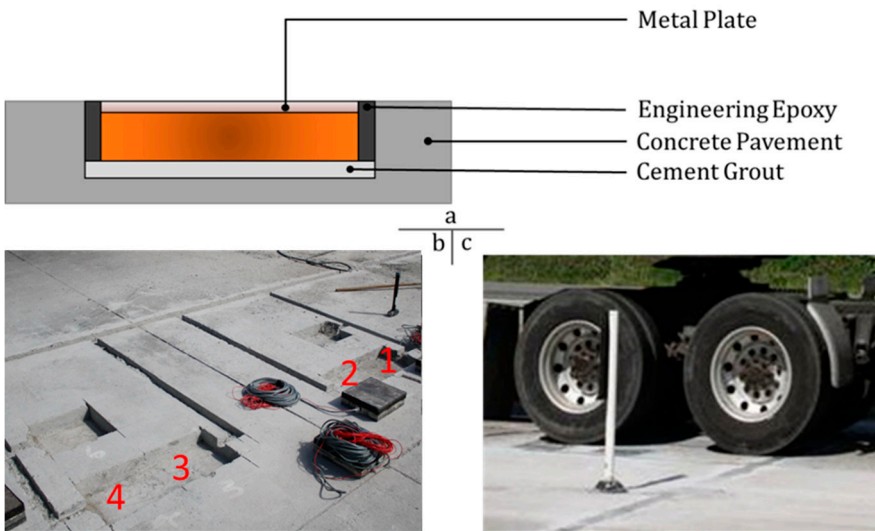

**Figure 4.** Installation of energy harvesting systems at I-81 Troutville weigh station (**a**) sketch of the side view; (**b**) the installation pits and P-WIMs; and (**c**) testing with real traffic).

*2.5. Data Acquisition*

The weigh station was open to traffic three days after the installation. However, the traffic speed is strictly controlled at about 40 km/hour. It makes it impossible to perform the evaluation at different vehicle speeds. The electrical outputs from the installed P-WIMs are measured using a National Instrument digital multimeter USB-4065 (Manufactured by Natinal Instrument, Austin, Texas, United States) connecting to a personal computer. The multimeter measures the characteristic voltage

generated from the P-WIMs, and the computer records the voltage over time. In each week, 60 sets of electrical outputs data from each P-WIM are measured.

## 3. Testing Results

During the on-site evaluation, it was noticed that not all vehicles would pass over the P-WIMs contacting their entire top surfaces due to the vehicle wandering. Having contact with part of the tire causes the drop of the electrical output generated from P-WIMs. As shown in Figure 4d, the engineering epoxy has a darker color than the pavement surface. Some drivers noticed it would steer rather than driving directly over the P-WIMs. While the actual contacting area is unknown, it is difficult to calculate the axle loading with the generated voltage. Therefore it is necessary to filter the data and keep only the voltage generated from full contacting. Since the axle loading of the trailer trucks' first axle is almost the same, the peak value of the voltage generated from the first axle is selected as the reference to filter the measured data. Figure 5 shows a sample voltage output from the installed P-WIM and an instruction of finding the reference peak voltage. As is known, the voltage generated from piezoelectric material always has the positive part and a negative part. The applied force is mostly characterized by the positive part. To present the electrical output concisely, the negative part of the electric output is removed in the analysis. The curve of the voltage output over time presents a typical axle configuration of 5-axles trailer truck which is widely used in the United States. The data was measured on 25 October 2012, from P-WIM #1. Firstly, the maximum values of the peak voltage output of the first axle are found from each set of data measured from each P-WIMs as the reference peak voltage value. Data measured from P-WIMs #1 to 4 are used in data analysis. The voltage data are excluded from the analysis if the peak value of the first axle is less than 85% of the reference peak voltage value.

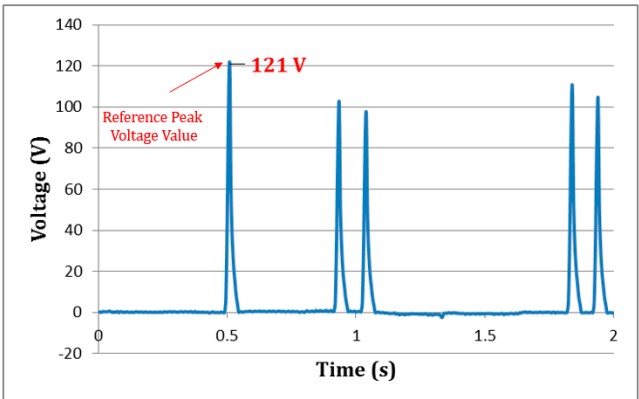

**Figure 5.** A sample data and instruction of finding the reference peak voltage value.

## 4. Discussion

### 4.1. Force Transmission

While transmitting the force from vehicles' wheels to the piezoelectric materials, some force is absorbed and wasted by the protective package and the surrounding pavement. Therefore the actual loading applied to the piezoelectric materials is less than the loading applied on the surface of the P-WIMs. Since the voltage generated by each disk is the same and all the piezoceramic disks are connected in parallel, the stress applied on each disk by the loading axle can be calculated as:

$$\sigma = V/(g_{33} \times t) \tag{1}$$

where $g_{33}$ is the voltage constant, and t is the thickness of the disks. The $g_{33}$ of the 850 series PZT material is 0.024 Vm/N and the thickness of the disks is 0.02m. The total loading actually taken by all 9 disks in the P-WIM can be calculated as:

$$F_i = \sigma \times \pi r^2 \times 9 \tag{2}$$

where r is the radius of the disks which is 0.02m, and i is the ID of the P-WIMs. The force transmission percentage, $P_T$, of the force applied from vehicles' axles can be calculated as:

$$P_T = \frac{\sum_{i=1}^{n=4} F_i}{F_0} = [\sum_{i=1}^{n=4} \frac{9 V_i \pi r^2}{(g_{33} \times t)}] / F_0 \tag{3}$$

where $V_i$ is the voltage measured from each of the installed P-WIM and $F_0$ is the vehicle axle loading provided by VDMV. All the data from VDMV was recorded with a time log. The time setting on the testing computer was reset to match the time setting used in VDMV (weigh station) computer. As an example, the axle loading of one trailer truck passed the weigh station is listed in Table 1. The peak voltage outputs from the P-WIMs corresponding to this trailer truck are listed in Table 2. Table 3 listed the calculated actual loading applied to the 9 disks in each P-WIM and the force transmission percentage. The average force transmission percentage for this measurement is about 15.16%. Thirty-three sets of data measured on 25 October 2012 are generated from full contact and valid to perform the analysis. The average force transmission percentage of them is about 14.84% with 0.63% of the variance and 0.79% of standard deviation.

**Table 1.** The axle loading data of a trailer truck passed the weigh station.

|  | Axle 1 | Axle 2 | Axle 3 | Axle 4 | Axle 5 |
|---|---|---|---|---|---|
| Axle Loading (kN) | 40.5 | 76.1 | 72.1 | 81.8 | 77.4 |

**Table 2.** The peak voltage outputs from the installed P-WIMs corresponding to the example trailer truck.

|  | Axle 1 | Axle 2 | Axle 3 | Axle 4 | Axle 5 |
|---|---|---|---|---|---|
| P-WIM 1 | 121 V | 100 V | 96 V | 109 V | 103 V |
| P-WIM 2 | 0 | 132 V | 126 V | 143 V | 135 V |
| P-WIM 3 | 0 | 111 V | 106 V | 121 V | 113 V |
| P-WIM 4 | 142 V | 142 V | 136 V | 155 V | 146 V |

**Table 3.** The actual loading applied to the piezoelectric materials in each P-WIM and the force transmission percentage.

|  | Axle 1 | Axle 2 | Axle 3 | Axle 4 | Axle 5 |
|---|---|---|---|---|---|
| F1 | 2,852 N | 2,361 N | 2,254 N | 2,570 N | 2,416 N |
| F2 | 0 | 3,103 N | 2,962 N | 3,378 N | 3,175 N |
| F3 | 0 | 2,612 N | 2,493 N | 2,843 N | 27,672 N |
| F4 | 3,348 N | 3,354 N | 3,201 N | 3,650 N | 3,431 N |
| Total | 6,201 N | 11,430 N | 10,909 N | 12,442 N | 11,695 N |
| Axle Loading from VDMV | 40,500 N | 76,100 N | 72,100 N | 81,800 N | 77,400 |
| PT | 15.31% | 15.02% | 15.13% | 15.21% | 15.11% |

## 4.2. Cost-Effectiveness Analysis

The main objective of this study is to provide a low-cost alternative to conventional permanent WIM system. The cost of fabricating one proposed P-WIM and the total cost of installing P-WIMs to cover a two-lane road is compared to a typical permanent WIM. A P-WIM consists of several parts, including nine pieces of 850 series piezoelectric material disks, the protective package made from

engineering plastic and copper alloy electrodes. Additional testing tools are also required for axle loading measurements, such as a National Instrument digital multimeter and a personal computer. According to the research report of MnDOT published in 2016, the cost of conventional four-lane Lineas sensors and a commercial controller used by MnDOT is estimated at $133,072 [31]. The detailed comparison between the cost of P-WIM and a permanent WIM for four lanes are listed in Table 4. The total cost, including the system and installation of P-WIM, is 80% less than the Permanent WIM system used by MnDOT.

**Table 4.** The comparison of the cost of P-WIM and regular WIM systems.

| | | P-WIM | | | Permanent WIM | | | |
|---|---|---|---|---|---|---|---|---|
| | | Unit Price | Quantity | Total | | Unit Price | Quantity | Total |
| **Sensing** | 850 series Piezoelectric Disks | $25 | 144 | $3,600 | Quartz Lineas sensor with 100 m lead cable | $6,232 | 16 | $99,712 |
| | Engineering Plastic Protective Package | $200 | 16 | $3,200 | Roadtrax BL sensor with 400 ft lead cable | $800 | 8 | $6,400 |
| | Copper Alloy Electrodes | $10 | 16 | $160 | | | | |
| **Measuring** | Power Cables | $25 | 16 | $400 | Controller | $26,000 | 1 | $26,000 |
| | Digital Multimeter | $300 | 1 | $300 | | | | |
| | Personal Computer | $800 | 1 | $800 | | | | |
| **Installation** | Contractors & Equipment | | $13,000 | | Grouts | | $7,872 | |
| | Materials | | $2,000 | | | | | |
| **Total** | | | $23,460 | | | | $133,584 | |

## 5. Conclusions

In this work, a P-WIM system designed for measuring the axle loading of the passing vehicle was introduced as a low-cost alternative to the conventional WIM system. The P-WIM consists of several piezoelectric disks sealed in a protective package which is made from engineering plastic. The P-WIM is capable of generating characteristic voltage output from passing vehicles. The loading of each axle from a passing vehicle can be calculated by the characteristic voltage output and the force transmission percentage. The force transmission percentage should be either determined in the laboratory or on-site by a known loading such as a vehicle with known axle information. The total cost of using P-WIM to measure the traffic of a two-lane road is about $19,780, which is 80% less than using permanent WIM system.

Compared to permanent WIM, P-WIM still has many limitations. P-WIM requires the vehicle to pass it with the loading from its tires all applied to the top surface of P-WIM to provide accurate results of axle loading. During the on-site evaluation, more than 50% of the passing vehicles' axle loading cannot be measured. In the future, material filled in the installation pits and the material used for the protective package of the P-WIM should have a similar color to the local pavement to eliminate the visual difference. In addition, installing more P-WIMs would help capture more information of the passing vehicles. Another limitation of P-WIM is that the users have to determine the force transmission percentage of the installed P-WIMs since not 100% of the force from the wheel will be transmitted to the piezoelectric material. If the P-WIMs are installed with their top surface flush with the pavement surface, the force transmission percentage can be determined in the laboratory or on site. If the P-WIMs are installed under the pavement surface, the user has to determine the force transmission percentage before the installation. Either way, it requires the user to test the P-WIM with a vehicle with known axle loading or using loading applying equipment. The difference of force

transmission percentage of each P-WIM is likely caused by the immature manufacturing process (The volume of sealant, tolerance of the packaging material).

**Author Contributions:** Conceptualization, H.X.; methodology, H.X.; prototype manufacturing and testing, H.X. and Y.Z.; on-site evaluation, H.X. and Y.Z.; data collection and analysis, H.X.; writing—original draft preparation, H.X.; writing—review and editing, H.X. and Y.Z.

**Funding:** This research was funded by Federal Highway Administration, grant number BAA No. DTFH61-09-R-00017.

**Conflicts of Interest:** The authors declare no conflict of interest.

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
