# Peer review of "Feasibility Study for Using Piezoelectric-Based Weigh-In-Motion (WIM) System on Public Roadway"

_applsci, doi:10.3390/app9153098_

Round 1
Reviewer 1 Report
This study discusses the development of a piezoelectric material based WIM system. The study is of relevance in that WIM substitutes for permanent weighing stations that are expensive and placed typically on high volume roadways. The evaluation of the systems developed is satisfactory and shows a good comparison.
There is need to revise the literature to add recent research work in the area. Also, there is need to address the grammar and do a spell check of the writing.
Else a good paper and very relevant to the practitioner.
Author Response
First of all, the comments are appreciated by all the authors. Although there isn’t much progress on the in-pavement WIM in recent years, it’s necessary to present a research background in the manuscript. We have reviewed the studies of WIM technology conducted from 2017 to 2019 and concluded the results. The manuscript has also been carefully reviewed and all grammar mistakes are corrected.
Reviewer 2 Report
General comments:
Proofread! One typo error more or less at each line,
English language is mistreated,
WIM state-of-the-art is missing,
References are not adequate,
The idea of P-WIM is interesting but the scope of the article is not correct: the economical analysis is not valid for a system which is not working! But comparing technological solutions would be a good idea.
(Some) detailled comments:
Sentence lines 14 - 16 not clear,
Line 14: lack of, instead of lacking,
Line 20: Compared to...
Line 21: costS,
Line 31: the overloaded trucks have been.. I stop here the comments on typo errors and English language.
The introduction should give the background of WIM systems. More or less, the part on loading on rural roads could be made more precise and proved through references,
Lines 50 - 51: Cite a reference, because the contrary is generally said.
Line 68 - 70: References are irrelevant here.
Reference 20 not clear in the list of references (the whole proceedings?!),
Line 75: You should give some information about piezoceramic sensors (know to weigh badly),
Section 2.2 on material selection: cannot be understood. Explain the characteristics of piezoelectric materials (d, g, Q), before going in the explanations. By the way, you write d33 then d_{33].
Figure 3: give more information (slope, straight line, ..), see COST 323.
Figure 4b is of bad quality.
Lines 175 - 190: Generally the firsta xle load is used to recalibrate the data. Explain the consequence of using it to exclude data.
Equations 1 to 3 to be written better (mathematical expressions not displayed correctly),
Table 3: Explain the error (mean, variance) of your system.
Cost effectiveness analysis: you should know that there exist many WIM solutions, with as many prices.
Author Response
The comments from the reviewer are deeply appreciated by all the authors. It is very important for us to improve our manuscript. The following changes are made according to the comments:
1. The grammar mistakes are corrected by several times of proof readings.
2. The cost-effectiveness analysis is redone based on a four-lane road. Detail of the cost of existing WIM technology is added including corresponding references. The WIM technology we compared with is from MnDOT.
3. The information of average daily load on rural roads from 1970 to 2013 was concluded by Highway Statistics 2013 (only this year, they have not done similar research after 2013) by Federal Highway Administration. More detail is provided in the manuscript according to this part and the references are added.
4. The conclusion that “Installing permanent WIM system is not very cost effective since the installation cost can be several times more than repairing the road” was made based on the communication between us and Virginia DOT. This conclusion was a comparison of the maintenance cost (usually $45,000 per km) of rural roads and installing permanent WIM system on rural roads. However, after speaking to the VDOT again, we found that the maintenance cost highly depends on the severity of the failure or damage, which means there isn’t a certain number for the maintenance cost. Therefore, we found this conclusion is not very rigorous and we decided to remove it from the manuscript.
5. Reference 20 is corrected.
6. The references included in the manuscript are revised and now is more related to the research topic.
7. Figure 3 is revised and more detail is added to the figure.
8. Figure 4b is replaced with an enlarged image and the position for the four P-WIMs are marked with numbers.
9. Contents about using the piezoelectric materials (mostly piezoceramics) in in-pavement WIM technologies are added with citation.
10. The contents about material selection are revised with more detail of each parameter and how it contributes to load measuring.
11. Contents about the reason for excluding data is added. More detail about the data exclusion is added in the manuscript.
12. Equation 3 is rewritten to express the calculation more appropriately.
13. Contents expressing the variance of all 33 sets of data are added to the manuscript.
Reviewer 3 Report
Sec.2.5: Existing WIM systems are usually applied to some range of vehicle speed. Please describe the accuracy in a range of speed which is considered in service.
Sec.4. and Table 4 : The cost comparison in the table 4 seems to have some biased price for readers. If the described cost of “quartz type” is based on some existing WIM station or a specific (but conventional) WIM facilities in the US, please describe more information.
Author Response
The comments from the reviewer are very valuable to us to improve our manuscript.
The P-WIMs are designed to measure the vehicles’ axle loading based on the deformation of the piezoelectric materials sealed in the P-WIMs and generated voltages. Theoretically, the vehicle speed would not affect the measurement accuracy. The vehicle speed could also be calculated from the time interval between the voltage peaks. Since during the on-site evaluation, the vehicle speed was controlled, how the vehicle speed influence the measurement accuracy will be investigated in the future work. Such content is also added in the manuscript.
More detail about the cost of existing WIM technology is added including the corresponding reference. The WIM technology we compared to is from MnDOT.
Round 2
Reviewer 2 Report
Dear authors,
Thank you for your corrections.